# The Content and Emission form of Volatile Organic Compounds from Cooking Oils: A Gas Chromatography-Mass Spectrometry (GC-MS) Analysis

**DOI:** 10.3390/ijerph20031796

**Published:** 2023-01-18

**Authors:** Ge Zhang, Fulu Sun, Haichao Li, Yuanxin Lin, Kai Zhao, Lei Fang

**Affiliations:** 1School of Civil and Resource Engineering, University of Science and Technology Beijing, Beijing 100083, China; 2Department of Environmental and Resource Engineering, Technical University of Denmark, 2800 Kongens Lyngby, Denmark

**Keywords:** cooking oil fume, VOCs, distribution form, GC-MS

## Abstract

Cooking oil fumes are full of dangerous chemicals that are bad for human health. Volatile organic compounds (VOCs) in cooking oil fumes are not only emitted in the form of gas but may also accumulate with other substances in oil fumes and form particulate matter emitted into the atmosphere. Different forms of VOCs can enter different regions of the human body and have varying effects on health. This paper investigated the VOC emission types found in some cooking fumes. The findings demonstrate that organic contaminants from edible oils were released as gas and particle matter, with gas being the predominant component. The fraction of gaseous VOCs steadily declined as oil temperature rose, whereas the proportion of VOCs released as particulate matter gradually rose. It is possible to assume that the increase in oil fume with temperature was caused by the original oil’s components volatilizing more frequently under the influence of vapor pressure and that chemical reactions were not the primary cause of oil fume creation.

## 1. Introduction

The fumes emitted from heating vegetable oils contain a large amount of volatile organic compounds (VOCs), some of which have been linked to cancer, including polycyclic aromatic hydrocarbons (PAHs), aromatic amines, nitro-polycyclic aromatic hydrocarbons, etc. [1,2,3,4,5,6]. Long-term exposure to cooking emissions was linked to kidney damage [7], diseases of female reproductive organs [8,9], decreased lung functions [10,11], and also lung cancer among non-smoker women [12,13,14], according to epidemiological research. VOCs from cooking oils may be released as gases or particles [15,16], which can have a variety of health effects on different regions of the body [17]. In order to comprehend the impact of cooking fumes on human health and to build the groundwork for future studies on VOC exposure in the human body, it is necessary to analyze the components of VOCs in various emission forms.

Cooking methods are indicated by the temperature of the oil, and as the temperature of the oil rises, so does the rate of VOC emission [18]. When See and Balasubramanian [19] investigated four different cooking techniques, they discovered that deep-frying produced the most chemical components and Particulate Matter 2.5 (PM2.5), followed by stir-frying, boiling, and steaming. According to Kabir and Kim’s research [20], stir-frying emits 3–5 times as many volatile organic compounds (VOCs) as deep-frying.

Another key element influencing the emission of VOCs is the kind of oil. Based on GC-MS, He et al. [21] investigated the chemical makeup of VOCs released by vapors from cooking oil. They discovered that aldehydes, ketones, and alkanes dominated the gas-phase VOCs released from five common vegetable oils. The sequence of the VOC concentrations at 260 °C is olive oil > peanut oil > sunflower oil > soybean oil > blending oil. Fatty acids were discovered to be the primary constituents of particle-phase VOCs released from various vegetable oils in the research of particle-phase VOC composition [22,23,24]. Different vegetable oils generate particle-phase VOCs in the following order: olive > peanut > canola > soybean > sunflower [25,26].

All known studies have evaluated the VOC emissions from cooking oils either in the gas phase or the particle phase, while the phase distribution of VOCs under various conditions has not yet been investigated. The emission types of three common vegetable oils (canola, sunflower, and corn oils) during the heating process were studied in this study. Controlling the oil temperature with relevant literature allowed for the simulation of the steaming, frying, and deep-frying processes. The distribution of VOC emission and the impact of oil product and oil temperature on the distribution of VOCs were investigated using GC-MS analysis of the collected oil fume samples.

Additionally, the health risks of the main cooking pollutants (benzene, dichloromethane, and trichloromethane) of various oils at various temperatures were evaluated. The findings, particularly the health risk analysis under different circumstances, will significantly advance the field of health assessment research.

## 2. Materials and Methods

### 2.1. Experimental Materials and Apparatus

#### 2.1.1. Experimental Materials

The following vegetable oils that are commonly consumed by citizens were chosen: pressed canola oil, pressed first-class sunflower seed oil, and corn germ oil.

#### 2.1.2. Experimental Apparatus

The main instruments include GC-MS (Thermo Scientific ISQ 1300, Waltham, MA, USA), a cascade impactor, a gas flowmeter (Alicat Scientific, Tucson, AZ, USA), a heating device (ceramic heating plate with temperature controller, relay, and stainless-steel oil cup), an electric muffle furnace (SX-G04133), and a 0.9 m × 0.7 m × 1.6 m fume hood. The schematic illustration of the experimental apparatus is shown in Figure 1.

### 2.2. Sampling and Analysis Method

#### 2.2.1. Sample Collection

Quartz filter paper was used in this experiment to collect PM10 and PM2.5 from cooking odors. Prior to sampling, the quartz filter paper was placed in a muffle furnace and burned for 4 h to eliminate organic materials. The muffle furnace was heated as follows: the temperature was maintained at 150 °C for one hour, then increased to 300 °C for two hours, and finally to 450 °C for one hour.

The sampling was performed in a fume hood. A ceramic heating plate with a temperature controller was used to heat 20 milliliters of vegetable oil to 130 °C, 190 °C, and 270 °C, respectively. The sample pump was turned on for air extraction after the oil temperature had stabilized. Particles larger than 10 μm and 2.5 μm were collected on quartz filter sheets in the cascade impactor for 15 min at a flow rate of 28.3 L/min. The oil fume was diverted partially into the gas-washing bottle after passing through the cascade impactor to absorb gas-phase VOCs with acetone. A flow meter was used to control the flow rate of the oil fume through the gas-washing bottle at 0.1 L/min. Parallel samples were made for each experimental state during the test.

#### 2.2.2. Sample Analysis

Organic matter in particles on quartz filter paper was extracted for 15 min using ultrasonic oscillation with a 5 mL acetone solution. A 1 mL supernatant solution sample was obtained and kept in the GC-MS injection container after shaking and standing still. The gas-phase VOC sample was obtained straight from the washing gas bottle (1 mL solution) and placed into the GC-MS injection bottle.

The temperature of the chromatographic column was raised from 40 °C to 250 °C at a rate of 10 °C/min during the GC-MS analysis and was sustained for 0.5 min (total time was 21.5 min). The scanning range for mass spectrometry was 45 *m*/*z* to 350 *m*/*z*. The gathered samples were examined one at a time using Thermo Xcalibur software (Thermo Scientific Xcalibur 4.0.27.10, Thermo Fisher Scientific Inc., Waltham, MA, USA). The retention time was used to determine the chromatographic peaks after determining the effective chromatographic peaks in each spectrogram by adjusting the parameters of peak morphologies.

VOCs in the oil fume were quantified using the area normalization method and toluene equivalent conversion [21]. The formula for calculating the concentration of VOCs components was:(1)$$c_{i}=\frac{n_{i}\times M}{V\times ET}\times \frac{273+T}{273+T_{0}}$$
where $c_{i}\;$ is the concentration of each component of VOCs ($\mu g/m^{3}$); $n_{i}$ is the molar number of each component of VOCs; *M* is the molar mass of toluene (92.14 μg/μmol); *V* is the sampling flow rate (L/min), taking as 28.3 for calculating the concentration of particulate matter and 0.1 for calculating the gas-phase VOCs. *ET* is the sampling time (15 min). *T* is the temperature of the oil fume, that is, the heating temperature of cooking oil (130 °C, 190 °C, and 270 °C, respectively). $T_{0}$ is the temperature at standard state (20 °C).

## 3. Results

### 3.1. Composition of Oil Fume Emission

The collected samples were analyzed one by one with the help of Thermo Xcalibur software. After identifying the effective chromatographic peaks in each spectrogram by setting the parameters of peak shapes, the chromatographic peaks were determined according to the retention time. Figure 2 showed n-hexadecanoic acid identified from gas-phase samples from canola oil heated at 130 °C with a retention time of 24.70 min. Compared with the NIST database, the matching degree was 60.62%. The major detected substances in the fumes of the three oils are shown in Table 1.

The detected components of oil fumes in this study were similar to those found in earlier research. For example, Fang et al. extracted the volatile components of fumes from different seed oils and found that they mainly contained aldehydes and alcohols [27]. Sun et al. collected and analyzed VOCs from home-cooked dishes in various regions and found that alkanes had a high content, ranging from 33% to 71% [28]. Peng et al. pointed out that kitchen smoke contains acids, aldehydes, alcohols, and polycyclic aromatic hydrocarbons [29].

### 3.2. Rate of Oil Fume Emission

The VOC emissions of edible oil depend on the kind and temperature of the oil. As shown in Figure 3, the three types of edible oils released VOC concentrations that fluctuated with oil temperature. The data ranges of data were shown in Figure 3. The fume emission of three oils increased by roughly 20–30% when the oil temperature rose from 130 °C to 270 °C. The VOCs content of the corn oil emission was the least compared to the other two types of edible oils, and it was also the least variable at different temperatures. According to this result, corn oil is more suitable for high-temperature cooking, such as stir-frying and deep-frying.

After a temperature increase, the fume emission may increase due to physical, chemical, or a mixture of both types of change. If it is due to physical changes, the vapor pressure of the various components in the liquid oil rises with temperature, increasing the emission. If it is due to chemical transformation, something new is formed. Unsaturated fatty acid triglyceride could decompose during the heating process, producing free fatty acid, free glycerol, glycerol monoester, and diglycerol [26]. Free fatty acids degrade at high temperatures, and the rate of decomposition increases as the temperature of the oil rises, accelerating the generation of VOCs [30].

### 3.3. Emitting Forms of Oil Fume

The influence of oil temperature on the distribution of VOCs emission form of three vegetable oils is shown in Figure 4. The VOC emission types of three vegetable oils were distributed consistently. The major form of VOC emission was the gas phase, accounting for 98–99% of total emissions, while the proportions of PM10 and PM2.5 were roughly equal. Furthermore, as the oil temperature increased from 130 °C to 190 °C, the distribution of VOCs emission form was nearly uniform. However, when the oil temperature was raised from 190 °C to 270 °C, the amount of gas-phase VOCs emitted by canola, sunflower, and corn oils was reduced by 0.9%, 0.6%, and 1.3%, respectively. Both the relative proportions of PM10 and PM2.5 rose, with PM2.5 growing at a somewhat faster rate than PM10.

The increase in granular fume at high temperatures could be due to the release and coagulation of more components with higher molecular weights and boiling points, or it could be due to the formation of new macromolecules, which must be determined by analyzing the specific components in fume.

### 3.4. Changes in the Component Ratio of Oil Fume

The boiling temperatures and volatilities of VOC components are complex. As a result, the distribution of VOC components emission form varies substantially with oil temperature. The composition and distribution of VOCs in three different types of cooking oil were investigated. The major components of cooking VOCs were discovered to be acids and alcohols. The other components, which included a few aldehydes, benzenes, and unidentified oxidation products, accounted for a minor amount. Figure 5 depicts fluctuations in the total quantities and relative proportions of alcohols, acids, and other compounds exhaled by the three vegetable oils at different oil temperatures. In general, acids were the predominant constituents of cooking VOCs, but the quantity and form distribution of acids in total VOCs vary with oil type and heating temperature.

As the principal pollutant in canola oil fume VOCs, the rate and form of acid release were obviously affected by temperature. The overall concentration of acid in VOCs from canola oil rose with the increasing oil temperature, from 72.1 mg/m^3^ at 130 °C to 88.4 mg/m^3^ at 270 °C. When the oil temperature was less than 190 °C, the amount of acid in PM10, PM2.5, and gas pollutants were less impacted by temperature fluctuations, which ranged from 31% to 33%, 39% to 40%, and 69% to 74%, respectively. When the temperature of the oil increased from 190 °C to 270 °C, the relative fraction of acids in PM10 and PM2.5 increased dramatically, reaching 71% and 74%, respectively, exceeding the amount of gas-phase pollutants (69%). The quantities of hexanoic acid, heptanoic acid, octanoic acid, and nonanoic acid, with boiling points of 205 °C, 223 °C, 250 °C, and 255 °C, respectively, in PM10 and PM2.5 collected at 270 °C were considerably greater than those collected at 130 °C and 190 °C, according to GC-MS analysis of oil fume. When the oil temperature did not approach the boiling temperatures of the alkanes, it was supposed that just a small portion of the alkanes collected with other components in the oil fume to form particles and were ejected into the atmosphere. When the oil temperature reached the boiling temperatures of alkanes, a high number of alkanes volatilized, increasing the possibility of collision between diverse compounds in the oil fume, resulting in a significant rise in the relative proportion of acids in PM10 and PM2.5.

Furthermore, temperature affected the concentration of alcohols in canola oil fume, but the ratio of alcohols in pollutants did not change correspondingly with temperature. The overall concentration of alcohols in the canola oil fume increased from 18.5 to 25.7 mg/m^3^ as the oil temperature increased from 130 °C to 270 °C, while the total concentration of other types increased from 7.0 to 13.4 mg/m^3^. When the oil temperature increased from 130 °C to 190 °C, the proportion of alcohols and other organic compounds in the three types of emission forms barely altered. The proportion of fatty acids increased significantly when the oil temperature climbed from 190 °C to 270 °C due to the volatilization and breakdown effect. As a result, the fraction of alcohols and other organic molecules was reduced.

Sunflower oil and corn oil’s VOC variations with temperature were comparable to those of canola oil. One distinction was that compared to canola oil (7–10%) and corn oil (8–9%), sunflower oil had a larger relative fraction of other chemicals in gas-phase pollutants (21–28%). The gas-phase VOCs released by sunflower oil may contain more dangerous substances since other products from the oxidative breakdown of vegetable oil contained small amounts of aldehydes, benzenes, and other harmful molecules.

PM2.5, PM10, and the gas phase of the fumes were taken as the determining factors, and the relative content of alcohol and acid of all three oils was compared through single-factor analysis of variance (ANOVA). The results are shown in Figure 6. It can be seen that the significance levels of alcohol and acid comparison between PM2.5 and PM10 are 0.16187 and 0.69848, respectively, indicating that the relative content of alcohol and acid in large and small particle size particles is basically the same. In addition, the significance levels of alcohol and acid comparison between PM2.5 and gas phase are 0.00244 and 0.00321, respectively, and the significance levels of alcohol and acid comparison between PM10 and gas phase are 0.00003 and 0.00066, respectively, all of which are less than 0.01, indicating that the content of alcohol and acid in gas phase fumes is significantly higher than in particle phase fumes. This suggests that various impurities in oil, including various non-volatile components, can also be emitted in the form of condensation with alcohol and acid components.

### 3.5. Carcinogenicity of Gaseous Oil Fume

According to the US Environmental Protection Agency’s (US EPA) list of toxic pollutants in the atmosphere, many toxic volatile organic compounds were detected in our experiment, primarily methylene chloride, trichloromethane, and benzene. As a result, long-term exposure to the population’s cooking fume environment increases the risk of a variety of cancers. According to the Risk Assessment Information System (RAIS), the carcinogenic toxic effects of pollutants in oil fumes are classified and their carcinogenic risks are assessed.

The following formula can be used to calculate lifetime cancer risk (LCR) [31]:(2)LCR=CDI×SF
where LCR is the lifetime cancer risk (dimensionless); SF is the carcinogenic slope factor of pollutants (kg/mg); and CDI is the chronic daily intake of carcinogenic pollutants (mg/kg).

The following formula is used to calculate CDI of toxic exposure:(3)$$CDI=\frac{C\times IR\times t\times EF\times ED}{BW\times AT}$$
where, C denotes the concentration of atmospheric pollutants obtained from monitoring results (mg/m^3^); IR denotes the adult respiration rate (1.3 m^3^/h); t denotes the daily exposure time (h); EF denotes the exposure frequency (250 day/year, upper limit); ED denotes the age of exposure (25 years); BW denotes the body weight (65 kg); AT denotes the age of cumulative effect (25 years for non-carcinogenic effect; 70 years for carcinogenic effect).

The LCR of dichloromethane, trichloromethane, and benzene in the three oils at various temperatures is shown in Figure 7. According to the information in the figure:

The LCR values of the three oils in gaseous VOCs all exceeded 10^−4^, indicating a high potential carcinogenic risk. The canola and corn oils both had LCRs that exceeded the limit value in particulate matter, whereas the sunflower oil had LCRs that were within the potential risk range (10^−6^–10^−4^).

The carcinogenic risk of various carcinogens in gaseous and particulate fume was not in the same order. Dichloromethane, trichloromethane, and benzene were the LCR sequence of gaseous VOCs and particulate matter in the canola oil. Trichloromethane had the highest LCR in gaseous VOCs of the sunflower oil, followed by dichloromethane and benzene. The particulate matter LCR order of the sunflower oil was benzene, trichloromethane, and dichloromethane. Dichloromethane had the largest LCR of the gaseous VOCs of the corn oil, followed by trichloromethane and benzene. The temperature had a significant impact on the LCR of corn oil particles, and there was no clear hierarchy among the carcinogens.

The LCR of benzene, trichloromethane, and dichloromethane in various cooking oil fume increased as oil temperature rose, with dichloromethane being more temperature-dependent. The LCR of methylene chloride did not exceed the limit when canola oil particles were heated to 130 °C, but it did so when the temperature was raised to 270 °C. The LCR of methylene chloride did not exceed the limit when corn oil particles were heated to 130 °C, but it did so when the temperature was raised to 190 °C.

The finding of carcinogenic VOCs was in line with earlier studies. For instance, Linxuan Li et al. examined the carcinogenic risk range of volatile organic compounds of canola in the state of particulate matter and demonstrated that benzene and dichloromethane were the primary ones [16]. J.E.C. Lerner et al. conducted a cancer risk study of volatile organic compounds trichloromethane and benzene in electromechanical maintenance and automotive painting centers and showed that the LCR of benzene and trichloromethane exceeded the limit value and had cancer risk [32]. Zhang Dingchao et al. studied the health risks of benzene and other substances in volatile organic compounds in commonly used oils (rapeseed, soybean, peanuts, corn, and lard), indicating that benzene has a high risk of cancer in cooking [33]. This study expands on earlier research by ranking the carcinogenic risks of various carcinogens in gaseous and particulate oil fumes in order to more thoroughly assess fume’s carcinogenicity.

## 4. Conclusions

The phase distribution, composition of each phase, and carcinogenic risk of oil fumes were studied through experimentation. It was found that the majority of the organic pollutants that were released during the heating of the edible oils were in the gaseous state. The amount of VOCs in gas steadily declined as oil temperature rose, whereas the amount of VOCs released as particulate matter gradually increased. VOCs in the gas phase and particle phase have different transmission abilities in the air. Therefore, compared to just analyzing the composition of oil fumes in previous studies, this study provides more valuable guidance for the VOC diffusion and health concern prediction of oil fumes. The results of this study were obtained from the procedure of heating oil. Food components may have varied effects during different cooking techniques; thus, further research is required since the phase analysis of the VOCs produced by oil is more complicated.

## Figures and Tables

**Figure 1 ijerph-20-01796-f001:**
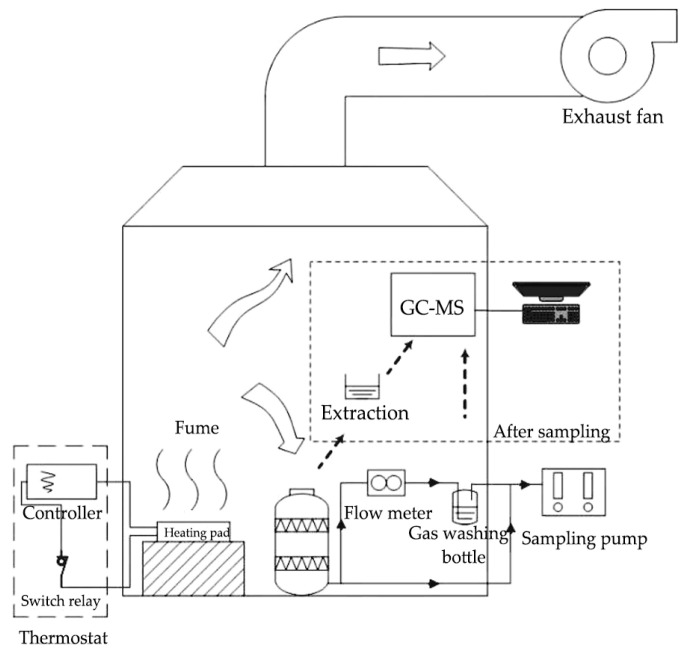
Schematic illustration of the experimental apparatus.

**Figure 2 ijerph-20-01796-f002:**
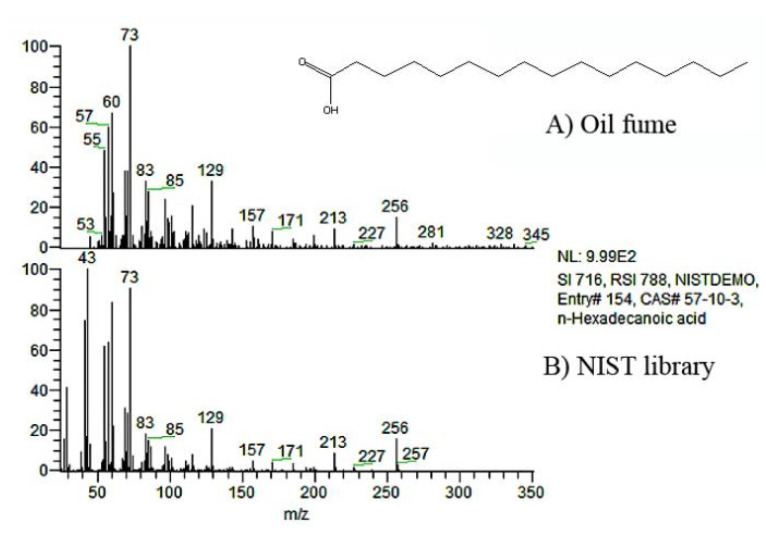
Mass spectrum of n-Hexadecanoic aide.

**Figure 3 ijerph-20-01796-f003:**
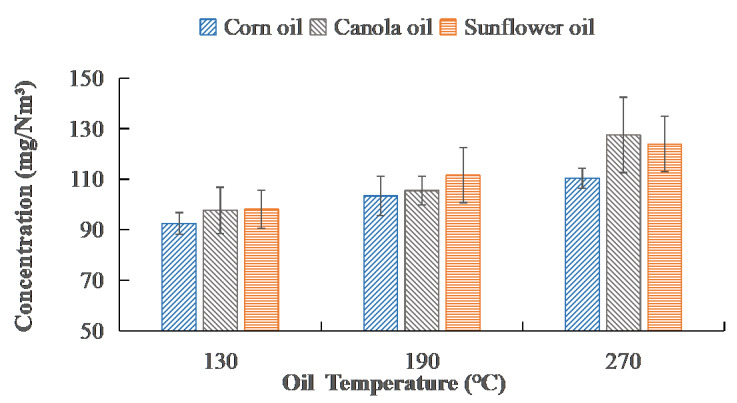
Emission concentrations of three vegetable oils at different temperatures.

**Figure 4 ijerph-20-01796-f004:**
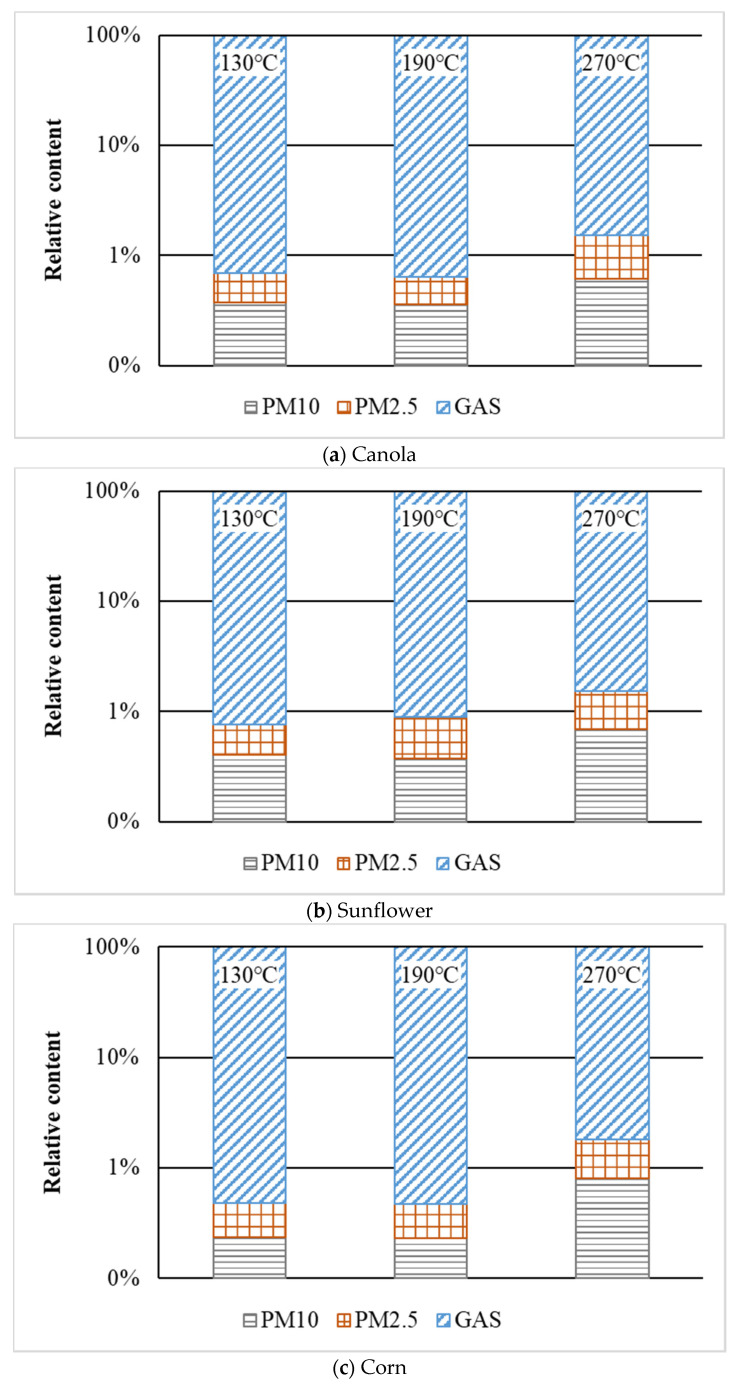
Distribution of VOCs emission forms at different temperatures.

**Figure 5 ijerph-20-01796-f005:**
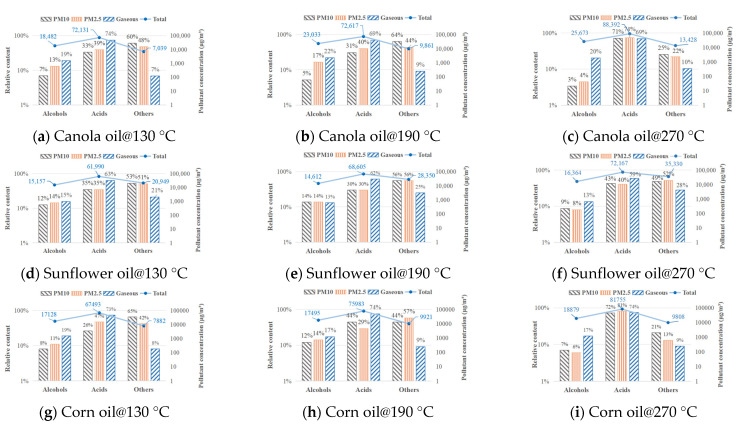
Emission rate and composition of VOCs.

**Figure 6 ijerph-20-01796-f006:**
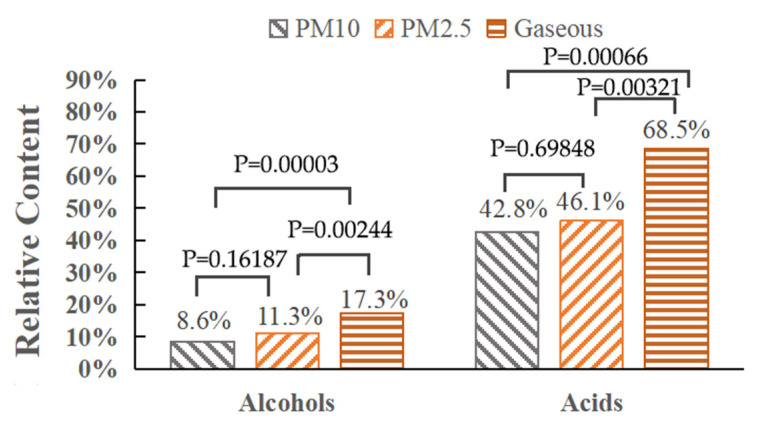
Result of single-factor analysis of variance.

**Figure 7 ijerph-20-01796-f007:**
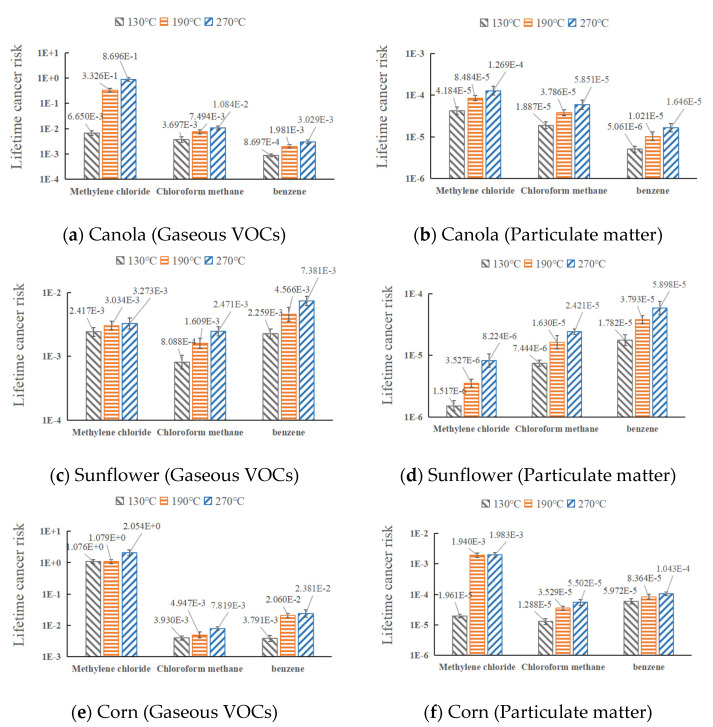
Cancer risk of selected pollutants in cooking fume.

**Table 1 ijerph-20-01796-t001:** Major detected substances.

Class	Name	Amount
Acid	n-Hexadecanoic acid	11
	Octanoic acid	
	undecanoic acid	
	Pentadecylic acid	
	Tridecanedioic acid	
	Octadecanoic acid	
	Hexanoic acid	
	Gibberellic acid	
	Ricinoleic acid	
	Hydrazinecarboxylic acid	
	11-bromoundecanoic acid	
Alcohol	Cyclopentadecanol	4
	Diacetone alcohol	
	Butoxy triglycol	
	Z-9-Hexadecen-1-ol	
Amine	Dextroamphetamine	4
	Butylamine	
	N-Ethylformamide	
	phenylethylamine	
Benzene series	Ethylbenzene	4
	Xylene	
	Toluene	
	Benzene	
Ester	Oleic acid methyl ester	3
	Hydroxy undecanoate lactone	
	Dodecalactone	
Halogenated hydrocarbon	Tetramethylsilane	3
	Trichloromethane	
	Dichloromethane	
Saccharide	Pentopyranose	2
	Trehalose	
Aldehyde	Digitoxin	2
	Vitamin A aldehyde	
Ether	Pentaethylene glycol monomethyl ether	2
	Hexaethylene glycol dimethyl ether	
Acid ester	Bufotalin	1
Phenol	3-methyl phenol	1
Alkanes	Undecane	1

## Data Availability

All data in this study are available from the corresponding author by request.

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
