# Peer review of "The Content and Emission form of Volatile Organic Compounds from Cooking Oils: A Gas Chromatography-Mass Spectrometry (GC-MS) Analysis"

_ijerph, 2023, doi:10.3390/ijerph20031796_

Round 1

Reviewer 1 Report

In the manuscript, Zhang et al. investigated the content and emission form of VOCs from three regularly consumed cooking oils by means of GC-MS. Unfortunately, several key points are missing in the context. The concerns are as follows,

1.     The authors stated that they applied GC-MS to characterize VOCs. However, I did not see any spectra figures and tables containing the molecules identified and quantified. Moreover, in the 2.2.2 sample analysis section, the authors should provide more information to justify that how did they identify molecules and how many times they tested per groups?

2.     The statistical analysis section is missing in the context. In fact, the authors just used percentages change instead of statistical approaches to compare differences among groups, which is unacceptable to come to conclusion.

3.     In the result and discussion section, limited space is for the discussion part. The authors just express the results of the study.

4.     Please re-write the conclusion section by avoiding repeating the results. Moreover, none of the data to demonstrate the “mechanism of oil fume production” as stated by the authors.  

Minor errors,

1.     Please substitute “VOCs” by “Volatile organic compounds” in the title.

2.     Line 55, one space more before “Additionally……”.

3.     Line 95, the “l” should be uppercase, such as “mL”.

Author Response

The reviewer provided us with a great deal of assistance to improve the work. This reviewer is very much appreciated. Answers to specific queries can be found below.

  1. The authors stated that they applied GC-MS to characterize VOCs. However, I did not see any spectra figures and tables containing the molecules identified and quantified. Moreover, in the 2.2.2 sample analysis section, the authors should provide more information to justify that how did they identify molecules and how many times they tested per groups?

Example of GC-MS result and major detected substances were supplemented as section 3.1.

  1. The statistical analysis section is missing in the context. In fact, the authors just used percentages change instead of statistical approaches to compare differences among groups, which is unacceptable to come to conclusion.

We agree with the reviewer that statistical analysis can better reflect the differences between groups. However, we think that the amount of data on the three oils is indeed not enough to do a non-parametric statistical analysis or even an analysis based on normal populations. In fact, we just listed the trend of change of experimental data, where the relative amount is not the relative amount between various groups, but the relative content of various forms of fume components in the same state. We hope the reviewer can agree with this processing, and we are willing to do further statistical analysis when there is enough data in the future.

  1. In the result and discussion section, limited space is for the discussion part. The authors just express the results of the study.

We recognize that the discussion is limited, so we change the title of the third part to "result".

  1. Please re-write the conclusion section by avoiding repeating the results. Moreover, none of the data to demonstrate the “mechanism of oil fume production” as stated by the authors.  

The statements including data which may repeat with the results were removed.

We agree with the reviewer and delete the statement about the mechanism.

  1. Please substitute “VOCs” by “Volatile organic compounds” in the title.

Revised.

  1. Line 55, one space more before “Additionally……”.

Revised.

  1. Line 95, the “l” should be uppercase, such as “mL”.

Revised.

Reviewer 2 Report

1- write the references according to MDPI style

2- add a table summarizing the molecular components of the emissions

3-  specify the impact of cooking method on the emission of VOCs (quick- and stir-frying, deep-frying cooking methods or mild cooking methods)

Author Response

We appreciate your careful review and inquiries. Answers to specific queries can be found here.

  1. write the references according to MDPI style

Revised.

  1. add a table summarizing the molecular components of the emissions

A table of the molecular components of the emissions was added in section 3.1.

  1. specify the impact of cooking method on the emission of VOCs (quick- and stir-frying, deep-frying cooking methods or mild cooking methods)

The impact to cooking was added in the first paragraph, section 3.2.

Reviewer 3 Report

The article is intresting and needed in the field.

Kindly take the below suggestions to improve the current manuscript. 

- TSD is suggested to be used with the different PM sizes instead of using PM10 and PM2.5, unless only PM10 and PM2.5 were measured. Please be specific with the accuracy of PM sizes that were measured. 

- Please write the full name then put the abbreviation is brackets; GC-MS.

- Line 65: The following vegetable oils that are regularly consumed by citizens were chosen: pressed canola oil, pressed first-class sunflower seed oil, and corn germ oil./ Based on what? Survey, reports,....

- Line 99: Please add the refrence that you followed.

- Results and discussion: the authors need to compare their results with the available reserach and justify their results. Refrences are needed to discuss the results.

Line 196: Kindly revise the caption

Line 280: Please recommend the areas that need furthur reserach.

Author Response

We appreciate your careful review and inquiries. Answers to specific queries can be found here.

  1. - TSD is suggested to be used with the different PM sizes instead of using PM10 and PM2.5, unless only PM10 and PM2.5 were measured. Please be specific with the accuracy of PM sizes that were measured. 

Only two levels of PMs were measured. As mentioned in the paper, “Particles larger than 10 μm and 2.5 μm were collected on quartz filter sheets in the cascade impactor”. It means PMs are just divided by such two filter sheets rather than referring to particles whose diameters are close to the two levels.

  1. - Please write the full name then put the abbreviation is brackets; GC-MS.

Revised.

  1. - Line 65: The following vegetable oils that are regularly consumed by citizens were chosen: pressed canola oil, pressed first-class sunflower seed oil, and corn germ oil./ Based on what? Survey, reports,....

We consulted the salesmen about the sales of oils in the market. The method should be survey. But we think “commonly” might be more accurate and substitute “regularly” with it.

  1. - Line 99: Please add the reference that you followed.

According to our knowledge, there is still no standards for measuring VOCs in particle oils. The extraction method adopted in the study was determined by our research experience.

  1. - Results and discussion: the authors need to compare their results with the available reserach and justify their results. Refrences are needed to discuss the results.

We added references in section 3.2.

  1. Line 196: Kindly revise the caption

Revised.

  1. Line 280: Please recommend the areas that need further research.

We believe that the presence of food will affect the formation of oil fume in the actual cooking process, which is worthy of further study. It was addressed at the end of the paper.

Reviewer 4 Report

The article "The Content and Emission form of VOCs from Cooking Oils: A GC-MS Analysis" is very interesting and written well. They have investigated emission of volatile organic compounds (VOCs) in cooking oil fumes and types of VOC emissions. Also they have investigated the relation between the temperature and release of VOCs and particulate matter in cooking oils. They have presented the article in a good format and described the experimental methods well. However I have few suggestions to improve the article, hence I recommend the article for a publication after a minor revision.

Please see the below suggestions. 

1. Line 36: Please expand PM ?

2. Introduction was written well, but please give the values of released VOCs (mentioned in line 43 and line 46) in the form of a table, it would be easy for the readers.

3. Experimental materials: Have you used any standards for detecting the VOCs and other particulate matter? and where did you obtained them? Please also describe the other materials used in this work and their source. 

4. Figure 1: Please improve the font, then it would be good.

5. Please cite corresponding reference articles for lines between 123-130. Similarly please cite the reference articles, when discussing about the results, through out the article

6. Figure 2: How many replicates have you used, please mention it in the article. Also do a significance test, then it would be clear, how significantly  are they differ? 

7. Figure 4: The visibility of this figure is not good. If possible present those results in the form of a table, it would be good.

8. Figure 4: Legend is not correct, please check the typo or errors. 

9. Figure 5: Also check the legend and correct it.

10. Author contributions are not mentioned, please do it.

11. Line 226, please check once.

12. Figure 5, how many replicates have you measured? and show error bars in the chart. 

Author Response

I would like to thank the reviewer for the careful review and constructive comments. Below are responses to specific questions.

  1. Line 36: Please expand PM ?

Revised.

  1. Introduction was written well, but please give the values of released VOCs (mentioned in line 43 and line 46) in the form of a table, it would be easy for the readers.

Not all literature data are given in tables, and it is difficult to reproduce accurate data when data is given in the form of figures in the original text. Would the reviewer allow the paper to retain qualitative ranking results and referring back to the text when precise values are needed?

  1. Experimental materials: Have you used any standards for detecting the VOCs and other particulate matter? and where did you obtained them? Please also describe the other materials used in this work and their source. 

According to our knowledge, there is still no standards for measuring VOCs in particle oils. The extraction method adopted in the study was determined by our research experience. GB18883-2022 Standards for indoor air quality (China) regulates the detecting method for indoor airborne VOCs, but we think the VOC in oil fumes might be too sticky to be collected solid sampling tubes, so that we use liquid adsorption method instead.

  1. Figure 1: Please improve the font, then it would be good.

Revised.

  1. Please cite corresponding reference articles for lines between 123-130. Similarly please cite the reference articles, when discussing about the results, through out the article

We added references in section 3.2.

  1. Figure 2: How many replicates have you used, please mention it in the article. Also do a significance test, then it would be clear, how significantly  are they differ? 

We took parallel sample during the test and gave the range of the data in Fig. 2. We did not conduct a significance test since the data was insufficient.

  1. Figure 4: The visibility of this figure is not good. If possible present those results in the form of a table, it would be good.

We will try to improve the visibility of the figure by editing, because we think it might be more understandable to readers. If we fail anyway, we will convert it to a table.

  1. Figure 4: Legend is not correct, please check the typo or errors. 

Revised.

  1. Figure 5: Also check the legend and correct it.

Revised.

  1. Author contributions are not mentioned, please do it.

Supplemented.

  1. Line 226, please check once.

Units were checked and modified.

  1. Figure 5, how many replicates have you measured? and show error bars in the chart. 

We took parallel sample during the test. Data ranges were added in figure 5.

Round 2

Reviewer 1 Report

Unfortunately, the authors did not convience me according to the revised version. In fact, all my important concerns are ignored, such as statistical analysis section and discussion section. In perticular, The authors just changed the title of the third section instead of extending the discussion section, which is unacceptable for me.

Author Response

  1. We agree with the reviewer's perspective that statistical analysis should be performed on data that meets the required sample size. In section 3.4, we presented data on the emission of various oils at different temperatures, using PM2.5, PM10, and gaseous forms of fumesas determining factors. Through single-factor variance analysis, we compared the relative content of alcohols and acids in each form and provided a figure (Figure 6) with levels of significance and interpretation.
  2. We also agree with the reviewer that it is necessary to include appropriate discussion in the results section. Our focus in the discussion is to emphasize the original value or consistency of the results with existing research. In section 3.1, we supplemented some literature and compared it with our detected fumecompounds to demonstrate the reliability of the experiment. In section 3.2, we analyzed the causes of fume generation and listed the literature on the decomposition of unsaturated and saturated fatty acids. Sections 3.3 and 3.4 are original parts of the paper, presenting the phase distribution of fume without comparative literature. In section 3.5, we analyzed the carcinogenic risk of fume and compared it with literature to demonstrate the rationality of carcinogenic risk analysis based on phase analysis. The above contents are all we can think of. We hope the reviewer can provide us with explicit instructions if additional content needs to be examined.
  3. We have rewritten the conclusion section. It summarizes the main points derived from the results of this study, points out the limitations of the research, and indicates the direction for further research.Due to out limited writing ability, the reviewer is welcome to correct any inappropriate aspects.

Round 3

Reviewer 1 Report

Thanks for the authors' efforts. This version would be suitable for publication.